# Multifunctional molecular hybrid for targeted colorectal cancer cells: Integrating doxorubicin, AS1411 aptamer, and T9/U4 ASO

**Kanpitcha Jiramitmongkon**[1,2,3], **Pichayanoot Rotkrua**[3,4], **Paisan Khanchaitit**[2], **Jiraporn Arunpanichlert**[1,3], **Boonchoy Soontornworajit**[1,3]*

1 Faculty of Science and Technology, Department of Chemistry, Thammasat University, Pathumthani, Thailand, 2 National Nanotechnology Center, National Science and Technology Development Agency, Thailand Science Park, Pathumthani, Thailand, 3 Thammasat University Research Unit in Innovation of Molecular Hybrid for Biomedical Application, Pathumthani, Thailand, 4 Faculty of Medicine, Department of Preclinical Science, Division of Biochemistry, Thammasat University, Pathumthani, Thailand

* sbooncho@tu.ac.th

**Data Availability Statement:** Minimal data sets are available in figshare repository: DOI 10.6084/m9. figshare.27924438.

## Abstract

Colorectal cancer (CRC) poses a global health challenge, with current treatments often harming both cancerous and normal cells. To improve efficacy, a multifunctional drug delivery platform has been developed, integrating bioactive materials, anticancer agents, and targeted recognition ligands into a single molecule. This study aimed to create a molecular hybrid (MH) containing doxorubicin, AS1411 aptamer, and T9/U4 ASO to regulate SW480 cell proliferation. The AS1411 aptamer targets nucleolin, overexpressed on cancer cell membranes, while T9/U4 ASO inhibits human telomerase RNA activity, further hindering cancer cell proliferation. AS-T9/U4_MH was synthesized via oligonucleotide hybridization, followed by doxorubicin loading and evaluation of its impact on cell proliferation. Binding capability of this MH was verified using fluorescence microscopy and flow cytometry, demonstrating specific recognition of SW480 cells due to nucleolin availability on the cell surface. These findings were corroborated by both microscopy and flow cytometry. AS-T9/U4_MH exhibited anti-proliferative effects, with the doxorubicin-loaded system demonstrating encapsulation and reduced toxicity. Moreover, the presence of Dox within AS-T9/U4_MH led to a notable reduction in hTERT and vimentin expression in SW480 cells. Additionally, examination of apoptotic pathways unveiled a marked decrease in Bcl-2 expression and a simultaneous increase in Bax expression in SW480 cells treated with Dox-loaded AS-T9/U4_MH, indicating its impact on promoting apoptosis. This molecular hybrid shows promise as a platform for integrating chemotherapeutic drugs with bioactive materials for cancer therapy.

## Introduction

To improve the efficacy of anti-cancer drugs, multifunctional drugs have been developed. These agents combine theranostic capabilities, functioning both as diagnostic tools and

**Funding:** This study was supported by Thailand Science Research and Innovation Fundamental Fund (Contract No. TUFF 27/2567), and Thammasat University Research Unit in Innovation of Molecular Hybrid for Biomedical Application. all funders had no role in study design, data collection and analysis, decision to publish, or preparation of the manuscript.

**Competing interests:** The authors have declared that no competing interests exist.

targeted drug delivery systems. By concentrating in target tissues and limiting distribution to other organs, they enhance treatment efficiency and reduce side effects [1]. Literature reports reveal the integration of a redox-responsive hyperbranched polymer with a nucleolin aptamer and doxorubicin to create a multifunctional drug delivery system. The aptamer acts as both a recognition element and a biological probe, while doxorubicin serves as the therapeutic agent for cancer. This innovative system has proven effective in mitigating severe side effects associated with cancer treatment and enhancing overall treatment efficiency [2]. The formation of multifunctional drugs employs a combination of noncovalent interactions and chemical strategies. These approaches have been successfully implemented in various forms. The outcomes of these strategies include soft-matter nanoarchitectures with defined sizes and morphologies, tunable luminescence, and specific biological functions [3]. One notable strategy involves the hybridization of complementary oligonucleotides to form complex structures in multifunctional drugs. This method allows for the integration of functional oligonucleotides such as antisense oligonucleotides (ASOs), small interfering RNAs (siRNAs), and therapeutic aptamers [4,5]. The current work proposes to expand on our understanding of multifunctional drug delivery systems, particularly those comprising aptamers and ASOs, by providing additional information and insights.

Aptamers are single-stranded DNA or RNA sequences identified via SELEX, known for high specificity and affinity. They offer advantages like easy modification, low immunogenicity, commercial availability, and versatile targeting [6]. The AS1411 aptamer, specifically, is a short single-stranded DNA with a 26-mer G-rich region. It serves as a recognition ligand for nucleolin, which is overexpressed on the surface of numerous cancer cells with remarkable selectivity and affinity. Moreover, the AS1411 aptamer, adopting a G-quadruplex conformation, has demonstrated anti-proliferation activity [7], increased resistance to nucleases [8] and enhanced cellular uptake [9]. Consequently, the AS1411 aptamer emerges as a promising candidate for use as a carrier for antisense oligonucleotides targeted at specific cancer cells.

ASOs are large nucleic acid molecules that selectively regulate the functions of a target gene. By virtue of their complementary nature to target mRNA, they engage in Watson-Crick base pairing, leading to the destabilization and degradation of the mRNA and, consequently, interference with the translation of the associated protein [10]. ASOs typically operate through two primary modes of action: splicing modulation and gene expression inhibition. In the role of splicing modulators, ASOs bind specifically to mRNA regions, blocking ribosome binding and thereby suppressing protein translation. As gene expression modulators, ASOs form duplexes with RNA, prompting RNAse H to degrade the targeted RNA sequences [11]. In existing literature, a specific ASO sequence known as T9/U4 has shown promising anticancer potential by inhibiting telomerase RNA [12]. Telomerase, a reverse transcriptase enzyme, comprises an RNA component (human telomerase RNA (hTR) or human telomerase RNA component (hTERC)) and a protein subunit (human telomerase reverse transcriptase (hTERT)) [13]. The proposed mechanism of action for this ASO involves direct inhibition of enzymes at the hTR active site, targeting the catalytic dysfunction of hTERT [14].

To suppress gene expression, the penetration of ASOs into target cells is essential. However, the precise mechanisms remain unclear, potentially involving factors such as size [15], structure [16], concentration, and cell line [17]. Challenges associated with ASO utilization include limited stability, non-specific delivery to target cells, and low cellular uptake, impeding their silencing effect [18]. Various delivery strategies, including viral vectors [19] and non-viral vectors [20] like liposomes, polymeric, and metal nanoparticles [21], have been developed to address these issues. Nevertheless, some of these systems are effective in cultured cells but face limitations in vivo biodistribution. Additionally, cationic lipids and polymers may induce adverse effects [22], while metal nanoparticles could accumulate in cells and tissues, leading to

mutation [23]. To overcome these challenges, molecular hybridization techniques have emerged as promising approaches for ASO and aptamer delivery. This method involves using ASOs with complementary sequences, exhibiting bioactivity, such as aptamers, to form double-stranded DNA and facilitate accumulation in target cells. Nucleic acid aptamers prove advantageous as intracellular delivery vehicles due to their specific and strong binding to cell surface receptor molecules [24]. Designed for versatile applications, these aptamers can be coupled with nanoparticles [25], drugs [26], and other nucleic acids [27]. Aptamers, serving as carriers for antisense oligonucleotides (ASOs), were developed with the goals of minimizing ASO dosage, mitigating off-target effects, enhancing cytocompatibility, improving selectivity, and optimizing cellular uptake [28].

Chemotherapy relies on anticancer drugs, but their lack of precision harms healthy cells alongside diseased ones. This highlights the need for more targeted therapeutic agents. Doxorubicin (Dox), an anthracycline drug, is commonly utilized in treating various cancers [29]. Despite its effectiveness, Dox has limitations due to side effects such as cardiotoxicity, nausea, vomiting, fatigue, alopecia, and oral sores [30]. To mitigate these adverse effects, Dox is often used in conjunction with other medicinal herbs [31] and drugs [32] to reduce toxicity, especially in cardiac tissues. Notably, Dox's structural composition includes flat aromatic moieties that intercalate between DNA base pairings making it a suitable candidate for incorporation into molecular hybrids [33]. In this study, a molecular hybrid named Dox-loaded AS-T9/U4_MH, comprising doxorubicin, the AS1411 aptamer, and the T9/U4 antisense oligonucleotide (ASO), was prepared, characterized, and assessed for its anti-cancer activities (Fig 1). The investigation focused on the primary adenocarcinoma colorectal cancer cell lines SW480 and Caco-2, along with the human normal colon cell line CCD 841 CoN, to evaluate the activities of the proposed molecular hybrid.

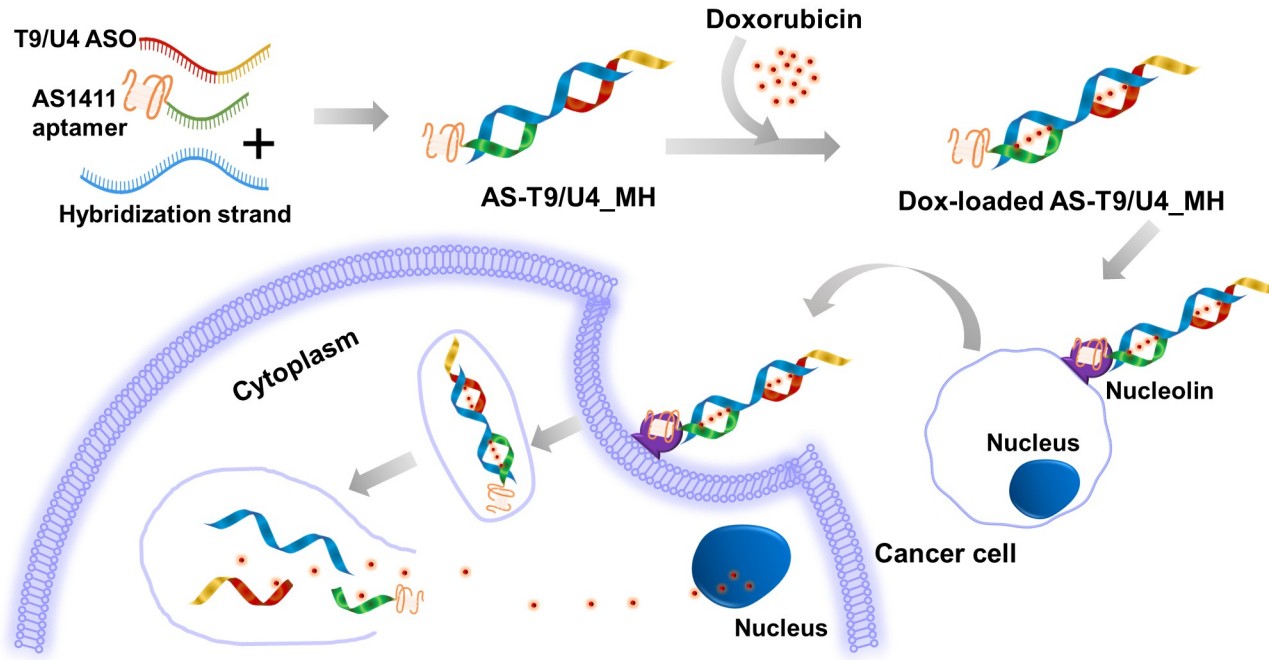

**Fig 1. The concept for preparing Dox-loaded AS-T9/U4_MH.**

**Table 1. The DNA sequences used in this study.**

| Name | Sequence (from 5′ to 3′) |
| --- | --- |
| Hybridization strand (HBS) | GAG TAT CCG TGT AAT GTG CTG ACA GAT CGA GCT TCG ATA GCC GAT |
| non-AS1411-HBS | TTC CTC CTC CTC CTT CTC CTC CTC CT CC AT CGG CTA TCG AAG CTC GAT |
| AS1411-HBS | GGT GGT GGT GGT TGT GGT GGT GGT GG CC AT CGG CTA TCG AAG CTC GAT |
| T9/U4-HBS | AGC ACA TTA CAC GGA TAC TC CC TAG GGT TAG ACA A |
| FAM-T9/U4-HBS | FAM – AAG CAC ATT ACA CGG ATA CTC CC TAG GGT TAG ACA A |
| non-T9/U4-HBS | AGC ACA TTA CAC GGA TAC TC CC ATT ACA GTG AGA G |

# Materials and methods

## Reagent

All the DNA molecules listed in Table 1 were purchased from Integrated DNA Technologies (IDT). Acrylamide/bis-acrylamide, tris-borate-EDTA buffer, ammonium persulfate, and gel loading buffer were obtained from Sigma-Aldrich. *N,N,N′,N′*- Tetramethylethylenediamine (TEMED) was sourced from Bio-Rad Laboratories.

## Preparation of a molecular hybrid comprising the AS1411 aptamer and T9/U4 ASO

To form a molecular hybrid, T9/U4-HBS, AS1411-HBS, and a hybridization strand with designated complementary sequences were combined in a PBS solution at a final concentration of 10 μM. The mixture was then incubated for 24 hours at room temperature to facilitate the hybridization of these three sequences. This resulting molecular hybrid was designated as AS-T9/U4_MH and confirmed through polyacrylamide gel electrophoresis. Additionally, two control molecular hybrids were prepared by substituting AS1411 aptamer and T9/U4 ASO with two non-specific sequences, denoted as nonAS-T9/U4_MH and AS-nonT9/U4_MH, respectively.

## Cell culture

Human colorectal adenocarcinoma cell lines (SW480 and Caco-2) and human normal colon cells (CCD 841 CoN) were obtained from the American Type Culture Collection (ATCC, USA) and cultured in DMEM medium supplemented with 10% fetal bovine serum, 1% penicillin-streptomycin at 37°C in a 5% $CO_2$ atmosphere. Additionally, the medium for Caco-2 cells was supplemented with non-essential amino acids. Cells were sub-cultured upon reaching 70–80% confluence.

## Evaluation of stability and binding of MH in vitro

The T9/U4-HBS sequences were labeled with carboxyfluorescein (FAM) at the 5' end prior to the formation of MH, allowing for the detection of fluorescence signals under a microscope and flow cytometer. The receptor-cellular binding of MH, mediated by the AS1411 aptamer, was assessed through microscopy and flow cytometry.

**Microscopic analysis.** SW480, Caco-2 (cancer cells), and CCD 841 CoN (normal cells) were seeded at a density of $9x10^5$ cells per well in 8-well chamber slide and incubated for 24 hours. Subsequently, the cells were treated with 10 μM of FAM-labeled nonAS-T9/U4_MH and FAM-labeled AS-T9/U4_MH for 1.5 hours. After treatment, the cells were washed twice

with PBS and stained with 4',6'-diamidino-2-phenylidoledihydrochloride (DAPI) to visualize nuclei. The cells were then imaged using a confocal laser scanning microscope (CLSM, ZEISS CLS 900).

**Flow cytometry.** SW480, Caco-2, and CCD 841 CoN cells ($3\times10^6$ cells per well) were seeded into 12-well plates and incubated for 24 hours. Afterward, the cells were treated with 5 µM of designated molecules, washed twice with PBS, trypsinized, collected, washed again with PBS, and resuspended in 0.5 mL PBS for binding analysis using flow cytometry.

## Cell proliferation assay

To assess cell proliferation, SW480, Caco-2, and CCD 841 CoN cells were seeded onto 96-well plates at a density of $5\times10^3$ cells per well and incubated for 24 hours. Subsequently, they were treated with various formulations including nonAS-T9/U4_MH, AS-nonT9/U4_MH, and AS-T9/U4_MH for 48 hours. Following treatment, MTS reagent (Celltiter 96®, Promega) was added, and the plates were further incubated for 1 hour before measuring absorbance at 490 nm using a microplate reader (Thermo Scientific, USA).

## Intercalation of Dox into MH

Incorporation of Dox into MH was achieved by incubating 10 µM of MHs with 0.95 µM of Dox at room temperature in the dark for 1.5 hours. The Dox concentration was applied based on its $IC_{50}$ value reported in a previous study [34]. The fluorescence intensity of Dox served as an indicator for intercalation, measured using a Varioskan microplate reader (Thermo Scientific, USA) with excitation wavelength set at 480 nm and emission wavelengths recorded from 500 to 800 nm. Dox-incorporated MHs were designated as Dox-loaded AS-T9/U4_MH, Dox-loaded AS-nonT9/U4_MH, and Dox-loaded nonAS-T9/U4_MH.

## Effect of Dox-loaded AS-T9/U4_MH on cell proliferation

The effect of Dox-loaded AS-T9/U4_MH on the proliferation of SW480, Caco-2, and CCD 841 CoN cells was investigated. Cells were seeded at a density of $5\times10^3$ cells per well in 96-well plates, cultured for 24 hours, and then treated with Dox, AS-T9/U4_MH, Dox-loaded non-AS-T9/U4_MH, Dox-loaded AS-nonT9/U4_MH, and Dox-loaded AS-T9/U4_MH, maintaining a concentration of 0.95 µM for Dox and 10 µM for MH for 48 h. Cell proliferation was determined using the MTS assay.

## Cell apoptosis

SW480 cells were seeded in 12-well plates at a density of $3\times10^6$ cells per well and incubated for 24 hours. Subsequently, cells were treated with Dox-loaded AS-T9/U4_MH, Dox-loaded AS-nonT9/U4_MH, AS-T9/U4_MH, AS-nonT9/U4_MH, and Dox for 48 hours at a final concentration of 0.95 µM for Dox and 10 µM for MHs. After treatment, cells were collected and subjected to apoptosis analysis using Annexin V-FITC/PI dual staining kit followed by flow cytometry.

## Western blot analysis

The cancer cells (SW480) were treated with 0.95 µM of free Dox, 10 µM of MH formulations: Dox-loaded AS-T9/U4_MH, Dox-loaded AS-nonT9/U4_MH, AS-T9/U4_MH, and AS-nonT9/U4_MH. After 48 hours of incubation, the cells were collected and lysed in RIPA buffer containing protease/phosphatase inhibitor on ice. Quantification of protein concentration was processed using the BCA protein assay (Thermo fisher). A total protein at concentration per

lane at 50 μg was loaded into the 10% SDS-PAGE and then transferred to PVDF membranes. Then, the membrane was incubated with blocking buffer (Intercept (TBS), Li-COR) at room temperature for 1 hour and then incubated overnight at 4˚C with the following primary antibodies: hTERT (cat. no. ab32020; 1:1000; 127 kDa; Abcam), vimentin (cat. no. ab92547; 1:1000; 54kDa; Abcam), β—actin (1:1000; 42 kDa, Cell signaling), Bcl–2 (1:1000; 25–28 kDa, Cell signaling), and Bax (1:1000; 20 kDa, Cell signaling). After incubation, the membrane was washed by 0.1% TBS—Tween-20. The second antibody, goat antirabbit IgG H&L/HRP antibody (AB6721, Abcam) were incubated on membranes (1:15000) for 1 hour at room temperature and then washed with 0.1% TBS—Tween-20. Subsequently, all bands were visualized using Odyssey XF imaging system (LI-COR). Densitometric analysis was used to evaluate the protein level, using β-actin as internal control.

## Statistical analysis

Statistical analysis was conducted using a one-way analysis of variance (ANOVA) test with a minimum of three experimental replicates to ensure statistical power. Results were presented as mean ± standard deviation (SD), with $P < 0.05$ considered statistically significant.

## Results and discussion

### Formation of AS-T9/U4 molecular hybrid

To confer the ability to downregulate human telomerase reverse transcriptase (hTERT) expression, we integrated an antisense oligonucleotide known as T9/U4 into our molecular hybrid, referred to as AS-T9/U4_MH. The T9/U4 sequence has been previously documented to effectively reduce hTERT expression [12,35]. The design of AS1411 aptamer (AS) and T9/U4 antisense oligonucleotide (T9/U4) sequences included additional oligonucleotides to facilitate hybridization with a hybridization strand (HBS). This was achieved by incubating HBS, T9/U4, and AS in a PBS solution at room temperature for 24 hours. The resulting product was characterized using gel electrophoresis. The gel image (Fig 2) demonstrated the successful formation of AS-T9/U4_MH. A distinct band approximately 90 base pairs in size indicated the presence of the molecular hybrid, as its size exceeded that of each individual building block: AS1411 aptamer, T9/U4, and HBS. Rotkrua et al. utilized a similar hybridization technique to form Chol-aptamer molecular hybrid (CAH) with high yield, as reported in their study [36].

### Stability and binding of MH *in vitro*

**Fluorescence microscope.** To demonstrate the potential of AS1411 aptamer as a carrier for T9/U4 to target tumor tissue, we utilized CLSM to observe Caco-2, SW480, and CCD 841 CoN cells after treating them with FAM-labeled MH (Fig 3). The fluorescence images revealed that FAM-labeled AS-T9/U4_MH specifically bound to SW480 cells. This specificity can be attributed to the known ability of AS1411 aptamer to bind to nucleolin, which is overexpressed on the surface of SW480 cells, as reported in the literature [36]. When AS1411 aptamer binds to nucleolin, it promotes the internalization of FAM-labeled AS-T9/U4_MH into the cytoplasm and nucleus via macropinocytosis, a common form of endocytosis in cancer cells [37]. This process is regulated by a nucleolin-dependent mechanism, which disrupts nucleolin-mediated trafficking and efflux. As a result, the MH-containing aptamer becomes trapped within cancer cells, leading to cell death [37,38]. Conversely, the fluorescence images of Caco-2 and CCD 841 CoN cells exhibited lower intensity, suggesting less binding of the FAM-labeled AS-T9/U4_MH. This discrepancy may arise from the lower availability of nucleolin in Caco-2 and CCD 841 cells. Research by Dean and Kenny has shown that the expression level

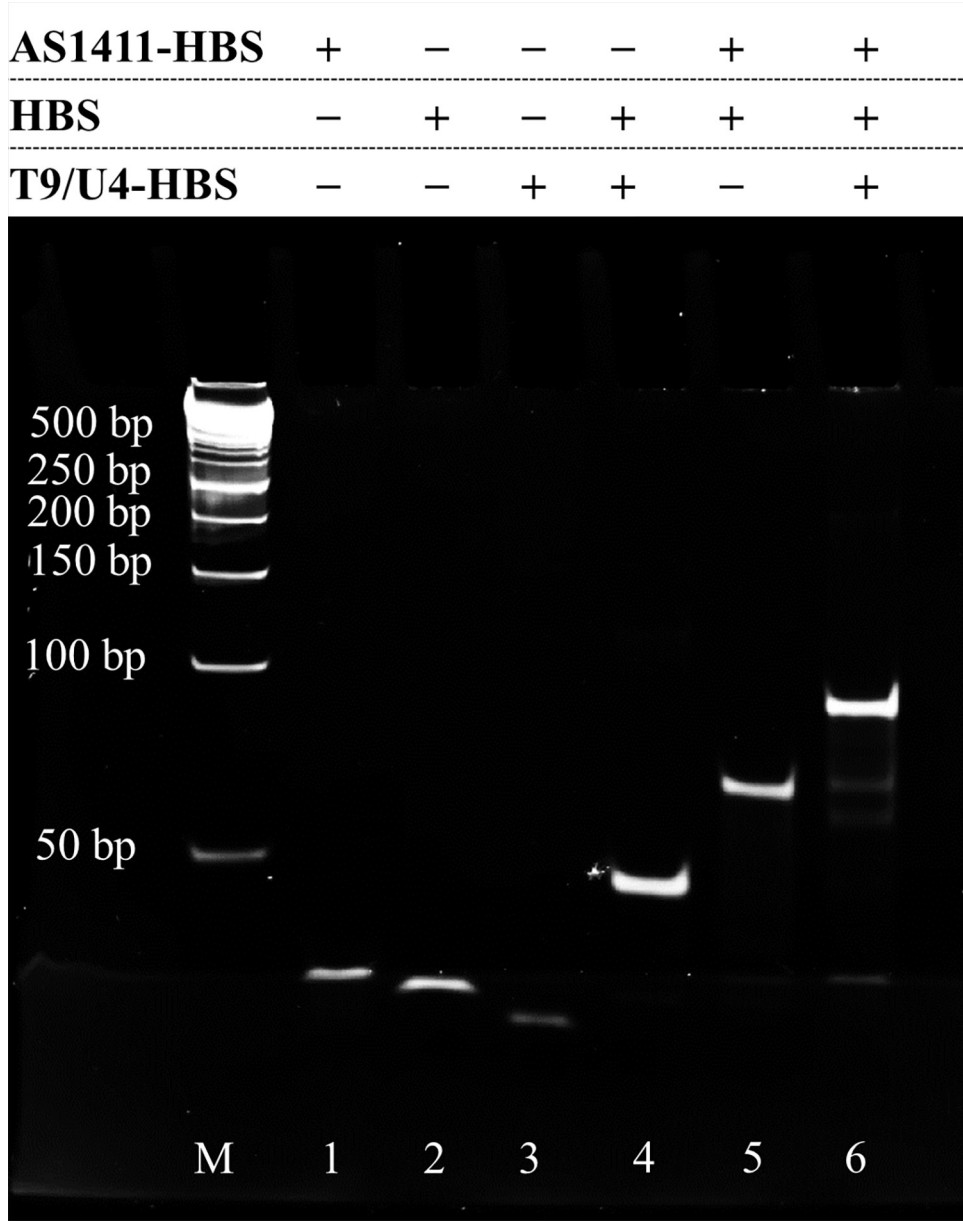

**Fig 2. Gel electrophoresis shows the assembly of AS-T9/U4_MH (lane M: DNA marker, lane 1: AS1411-HBS, lane 2: HBS, lane 3: T9/U4-HBS, lane 4: HBS + T9/U4-HBS, lane 5: HBS + AS-HBS, and lane 6: HBS + T9/U4-HBS + AS-HBS).**

of native nucleolin on the cell surface of intestinal Caco-2 cells is notably low, a finding corroborated by the detectability of this molecule using nucleolin antibodies such as MS-3 from Santa Cruz Biotechnology and ab22758 from Abcam [39,40]. Similarly, Lohlamoh et al. found that nucleolin mRNA expression in SW480 cells was significantly higher than in CCD 841 CoN cells [34]. Furthermore, Duncan et al. determined nucleolin's overexpression in both cancer cells and normal cells, including HS-27 (skin fibroblast), WI-38 (lung fibroblast), and MCF-10A (epithelial mammary cell). Their study highlighted that the nucleolin level in normal fibroblast cells was approximately four times lower than in fibroblast-like cancer cells such as HT-1080 and SK-MEL2 [41]. These findings collectively support the differential binding of

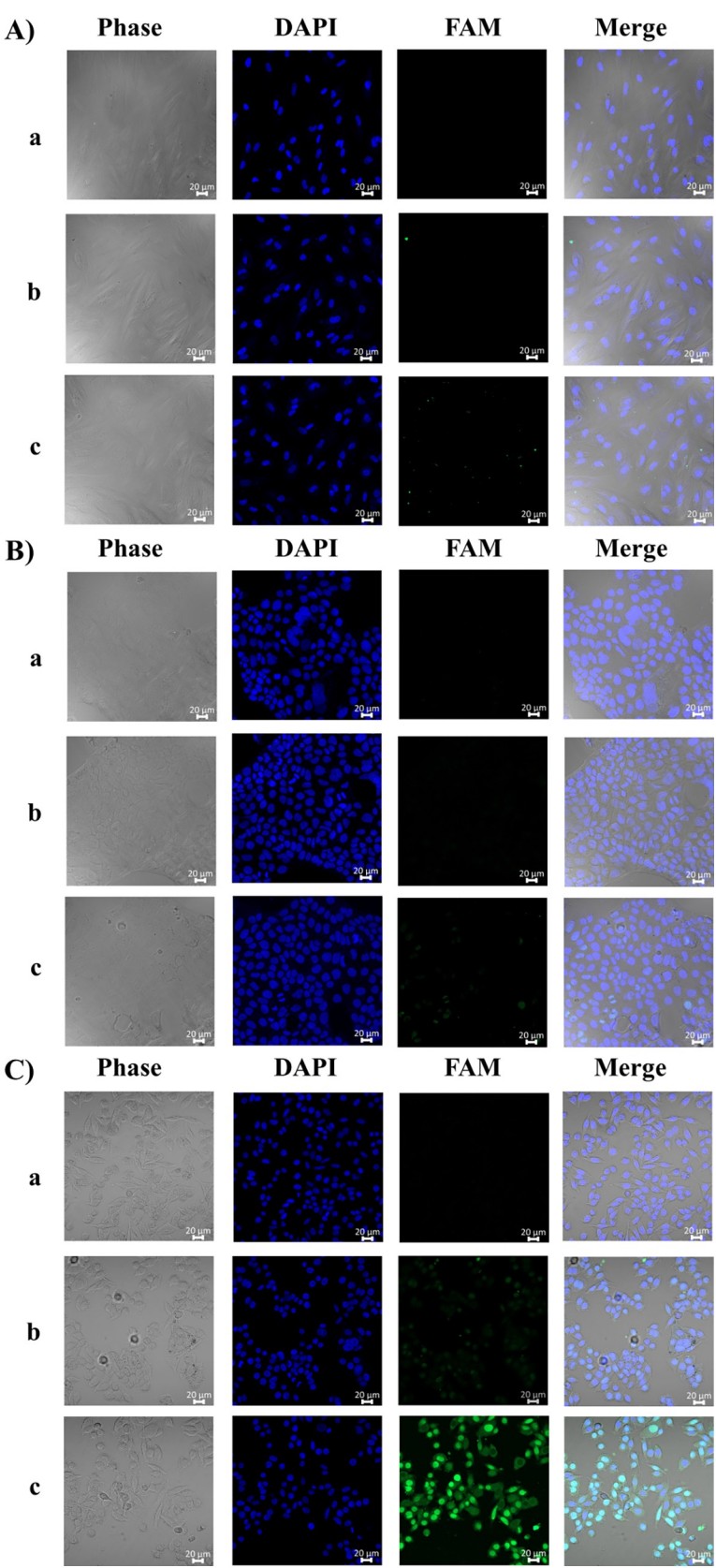

**Fig 3.** Fluorescence image of CCD 841 CoN (A), Caco-2 (B), and SW480 (C), treated with (a) no treatment, (b) FAM-labeled nonAS-T9/U4_MH, and (c) FAM-labeled AS-T9/U4_MH.

AS-T9/U4_MH to tumor cells versus normal cells based on nucleolin expression levels, reinforcing the potential of AS1411 aptamer as a targeted delivery vehicle for T9/U4 to tumor tissues.

**Flow cytometry.** Flow cytometry analysis was employed to evaluate the intracellular fluorescent intensity subsequent to the treatment of SW480, Caco-2, and CCD 841 CoN cells with FAM-labeled nonAS-T9/U4_MH and FAM-labeled AS-T9/U4_MH. FAM-labeled AS-T9/U4_MH treatment notably resulted in a significantly detected fluorescent intensity in SW480 cells compared to FAM-labeled nonAS-T9/U4_MH (Fig 4A and 4B). In contrast, Caco-2 and CCD 841 CoN cells showed minimal fluorescence differences between those treated with the MH formulation containing either the AS1411 aptamer or the non-binding sequence, as shown in the flow cytometry histogram (Fig 4A). This is likely due to non-specific binding between the tested cells and the applied formulation, a phenomenon frequently observed in biological assays [42]. These observations are consistent with our microscopy assay findings, further corroborating the preferential binding and uptake of AS-T9/U4_MH by SW480 cells in comparison to other cell types.

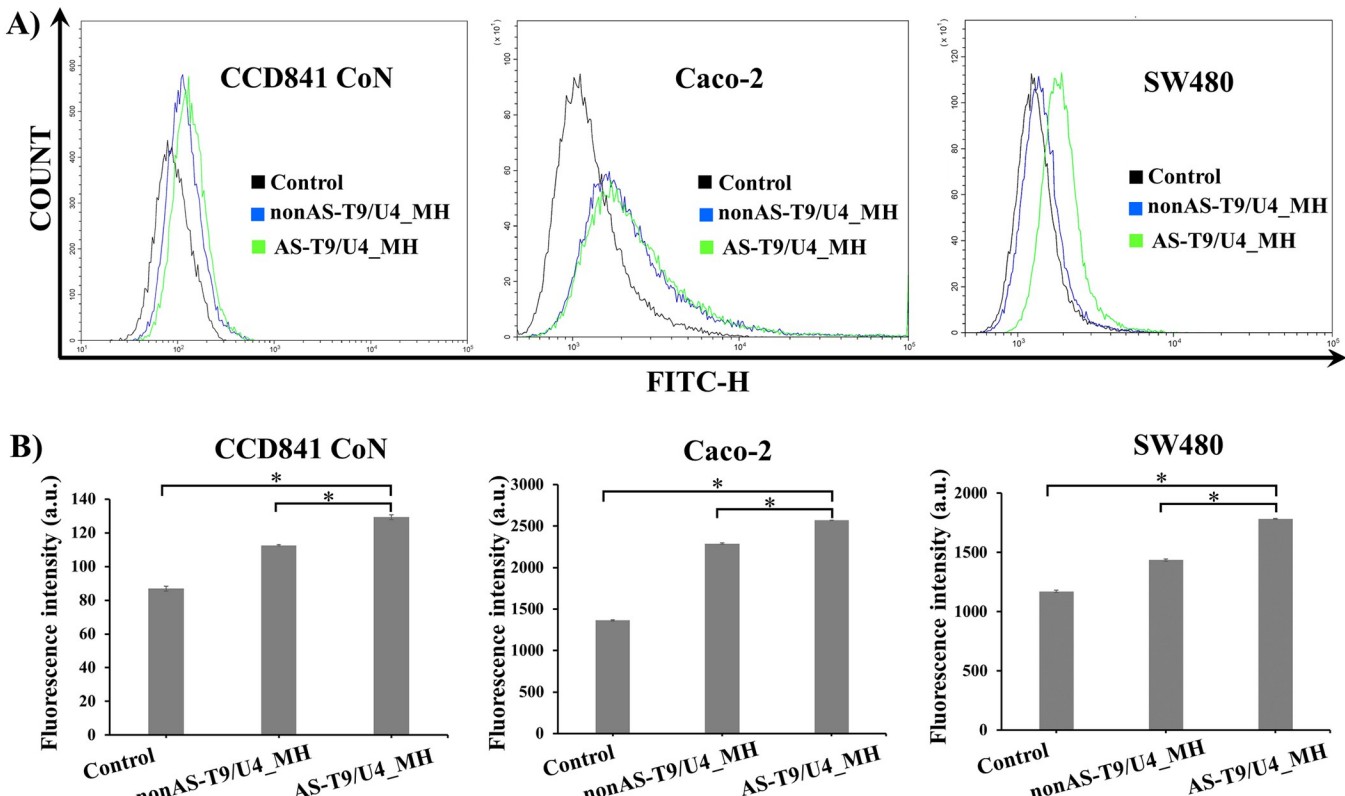

**Fig 4. AS-T9/U4_MH showed specific binding to SW480 cells.** A, B) Flow cytometry histogram of CCD 841 CoN (left), Caco-2 (middle) and SW480 (right). The black line represents cells without MH treatment. The blue line represents cells treated with FAM-labeled nonAS-T9/U4_MH. The green line represents cells treated with FAM-labeled AS-T9/U4_MH. The cells were treated at 37°C for 1.5 h, and no treatment as a control. *P < 0.05. The data are presented as means ± SD, n = 3.

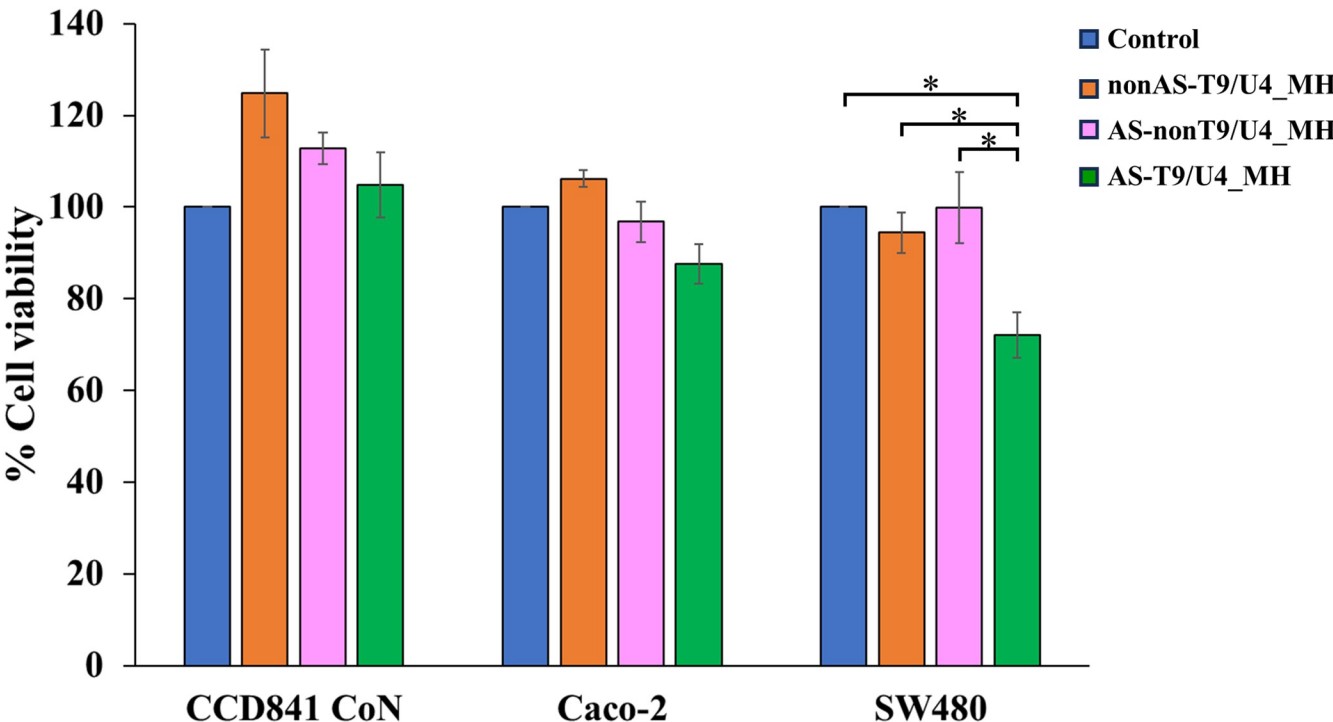

**Fig 5. The cell viability of CCD 841 CoN, Caco-2, and SW480 cells after treatments at 48 h of 10 μM nonAS-T9/U4_MH, AS-nonT9/U4_MH, AS-T9/U4_MH and no treatment as a control.** The values are presented as means ± SD, n = 3, *P < 0.05.

### Cell proliferation assay

To evaluate the effect of AS-T9/U4_MH on cell proliferation, we conducted MTS assays using SW480, Caco-2, and CCD 841 CoN cells. The results revealed that AS-T9/U4_MH exerted an anti-proliferative effect specifically on SW480 cells, while demonstrating no significant effect on Caco-2 and CCD 841 CoN cells (Fig 5). Notably, the cell viability of SW480 cells treated with AS-T9/U4_MH was lower compared to those treated with nonAS-T9/U4_MH. This specificity can be attributed to the presence of AS1411 aptamer within the molecular hybrid, as the aptamer is capable of recognizing nucleolin molecules present on the surface membrane of SW480 cells [43]. Research by Emilio et al. further supports the specificity of AS1411 aptamer, demonstrating its ability to significantly inhibit the phosphorylation of nucleolin in human umbilical vein endothelial cells (HUVEC), whereas control formulations such as scramble sequences and ranibizumab had no effect on nucleolin phosphorylation [44]. Conversely, AS-T9/U4_MH exhibited no discernible effect on Caco-2 and CCD 841 CoN cells, likely due to their lower nucleolin expression levels [41]. These findings underscore the targeted anti-proliferative effect of AS-T9/U4_MH on SW480 cells, driven by the specific interaction between AS1411 aptamer and nucleolin, highlighting its potential as a targeted therapeutic agent for tumors with elevated nucleolin expression levels.

### Intercalation of Dox into AS-T9/U4_MH

Fluorescence spectroscopy served as the method of choice to assess the intercalation of Dox into MH by detecting fluorescence quenching. In this work, Dox was excited using a light source at 480 nm, and the emitted fluorescence was detected with a maximum signal at 590 nm (Fig 6). The observed fluorescence quenching of Dox upon its incorporation into the

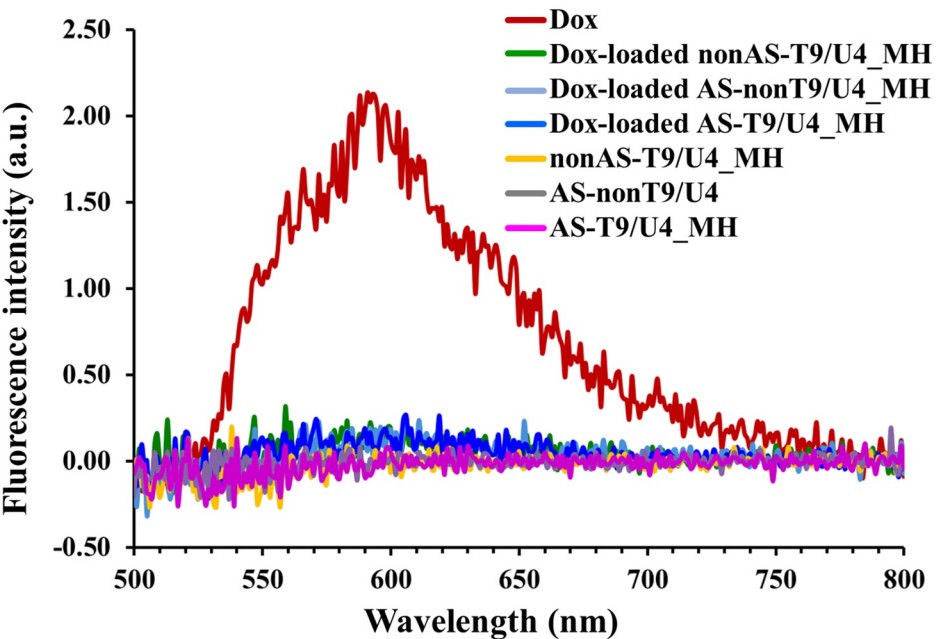

**Fig 6. Fluorescence spectra of 0.95 μM of Dox and 10 μM of Dox-loaded nonAS-T9/U4_MH, Dox-loaded AS-nonT9/U4_MH, Dox-loaded AS-T9/U4_MH, nonAS-T9/U4_MH, AS-nonT9/U4_MH, and AS-T9/U4_MH.**

molecular hybrids also demonstrated in S1 Fig indicates the complete intercalation of Dox within the double helix of DNA [45]. This phenomenon confirms the successful integration of Dox into the MH structure, validating its potential for targeted drug delivery. The fluorescence intensity of Dox was measured to evaluate drug loading into AS-T9/U4_MH using a calibration curve method (S2 and S3 Figs). The findings indicated that AS-T9/U4_MH could encapsulate approximately 98% of Dox within the complex. Additionally, the Dox loading capacity of AS-T9/U4_MH was assessed, as shown in S4 Fig, revealing that as the concentration of Dox increased, the fluorescence intensity also rose. When the molar ratio of AS-T9/U4_MH to Dox was 1:1, the fluorescence intensity of Dox nearly reached the minimum detectable signal, suggesting that the loading capacity of our MH was a 1:1 molar ratio. A Dox release study was performed by incubating Dox-loaded AS-T9/U4_MH in cell culture media at predetermined times and then collecting the media for absorbance measurement (S5 Fig). The result was calculated at the maximum absorption peak of 409 nm, which corresponded to a protonated tautomeric form of Dox in acidic solution [46]. At 48 and 72 h, Dox release from the complex reached up to 80% and 85%, respectively. The Dox release experiment demonstrated that only about 5% of Dox was released from MH after 1 hour, which aligns with the cellular imaging (S1 Fig). This minimal amount of Dox is likely insufficient to be detected inside the cells using fluorescence microscopy. In addition, the stability of MH in the cell culture environment was assessed. At specified time points, the released solutions were analyzed using gel electrophoresis. The gel image revealed a gradual decrease in the intensity of the band corresponding to MH as incubation time increased (S6 Fig), suggesting potential disintegration of the MH structure.

## Effect of Dox-loaded AS-T9/U4_MH on cell proliferation

Free Dox demonstrated toxicity towards Caco-2 and SW480 cells, while showing no impact on CCD 841 CoN cells (Fig 7), likely due to its therapeutic window being greater compared to

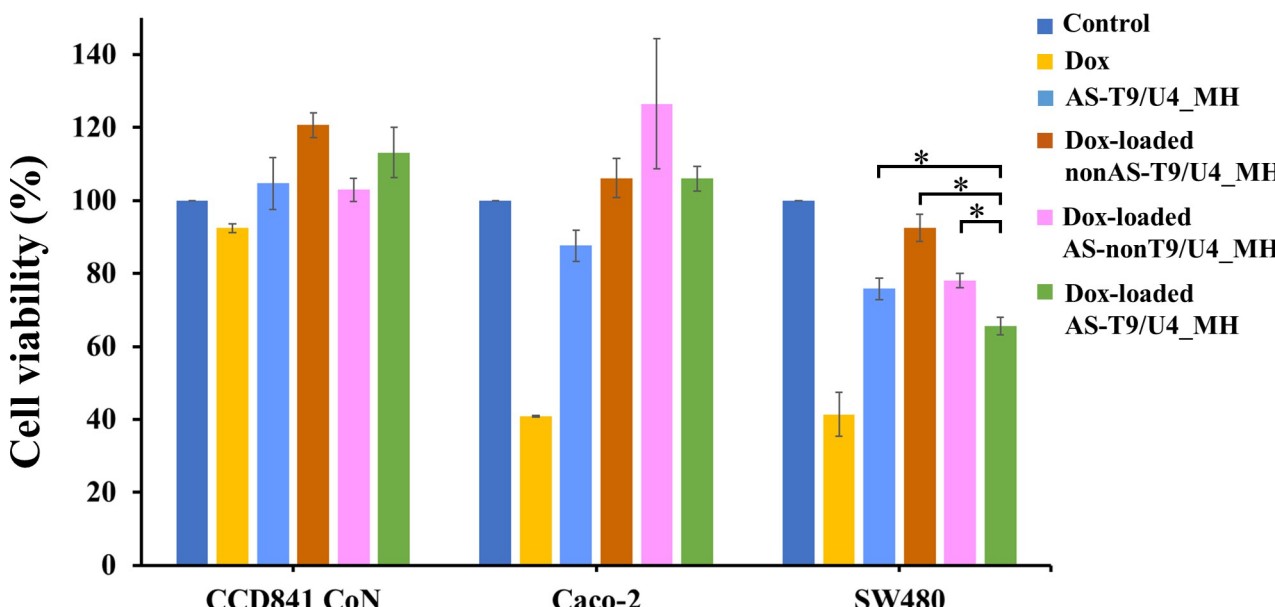

**Fig 7. The cell viability of CCD 841 CoN, Caco-2, and SW480 cells after treatments at 48 h of 0.95 μM Dox, 10 μM AS-T9/U4_MH, Dox-loaded nonAS-T9/U4_MH, Dox-load AS-nonT9/U4_MH, Dox-load AS-T9/U4_MH, and no treatment as a control.** The values are presented as means ±SD, n = 3, *P < 0.05.

Caco-2 and SW480 cells [36]. In the context of Caco-2 cells, upon intercalation of Dox into the DNA duplex of our molecular hybrids, a significant reduction in cytotoxicity was observed. This reduction can be attributed to the DNA duplex system's effectiveness in minimizing the cytotoxic effects of this anti-cancer drug, as previously reported in our study [36]. For SW480 cells, Dox-loaded AS-T9/U4_MH demonstrated the specific delivery of Dox to the cells when compared to the other two control molecular hybrids, as evidenced by the cell viability results. After cellular internalization, the Dox-loaded MH releases Dox molecules within SW480 cells, possibly driven by diffusion due to the relatively low intracellular Dox concentrations. Alternatively, the release may occur through the gradual degradation of the aptamer by lysosomal endonucleases following uptake. These mechanisms, as described in the literature, may work together to facilitate Dox dissociation [47]. We further analyzed and compared the cell viability levels in cells treated with AS-T9/U4_MH and Dox-loaded AS-T9/U4_MH. The results demonstrated a significant difference in cell viability between these two groups, highlighting the enhanced antiproliferative effect resulting from Dox incorporation in the MH. The enhanced effectiveness of this MH can be attributed to the synergistic effect resulting from the presence of both Dox and T9/U4 ASO. To further emphasize the efficacy of the MH in suppressing cell proliferation, a trypan blue assay was performed (S7 Fig). The results revealed that SW480 cells treated with Dox-loaded AS-T9/U4 exhibited significantly greater antiproliferative effects compared to AS-T9/U4, Dox-loaded nonAS-T9/U4, and Dox-loaded AS-nonT9/U4. These findings are consistent with the MTS assay results. The trypan blue assay provided a clear distinction between live and dead cells based on membrane integrity, further supporting the effectiveness of the MH. The result aligns with previous studies in the literature. For instance, Abaza et al. demonstrated that the antiproliferative effects of colorectal cancer cells were synergistically enhanced by combining c-myc antisense phosphorothioate oligonucleotides with anticancer drugs such as taxol, 5-FU, Dox, and vinblastine [48]. Similarly, Jhaveri et al. showed that combining human α isoform folate receptor (αhFR) antisense oligonucleotides with Dox

resulted in a five-fold reduction in αhFR expression in breast cancer cells [49]. The specificity of Dox-loaded AS-T9/U4_MH was achieved through the presence of AS1411 aptamer, as Caco-2 and CCD 841 CoN cells have lower availability of nucleolin [39,41]. Consistent with previous research, Dox intercalated into MH exhibited reduced toxicity compared to free Dox, suggesting that the MH strategy may offer a means to mitigate the adverse effects of Dox [36]. While our MH system showed reduced toxicity, its effective concentration was 10 μM, significantly lower than the reported $IC_{50}$ value of 144 μM for the AS1411 aptamer [50]. In comparison to other anticancer formulations based on aptamer-DNA nano-assemblies, which typically range between 3–30 μM [51,52], these findings suggest that our MH formulation holds promise as a potential cancer treatment with specific targeting characteristics and reduced toxicity.

## Effect of Dox-loaded AS-T9/U4_MH on cell apoptosis

Flow cytometry was employed to investigate the impact of T9/U4 ASO and Dox-loaded MHs on apoptosis in SW480 cells. The results revealed that the percentage of cell apoptosis was significantly higher in cells treated with AS-T9/U4_MH compared to those treated with AS-nonT9/U4_MH. This observation suggests that T9/U4 ASO effectively inhibited SW480 cell proliferation in vitro (Fig 8). Furthermore, the incorporation of Dox into AS-T9/U4_MH resulted in a more pronounced effect on apoptosis compared to Dox-loaded AS-nonT9/U4_MH, indicating a synergistic effect. This enhanced efficacy of Dox-loaded AS-T9/U4_MH indicates the potential of MHs as effective carriers for delivering therapeutic agents, such as Dox, in combination with ASOs for targeted cancer therapy. These findings show the promising synergistic effects of combining ASO-mediated inhibition of cell proliferation with the cytotoxic effects of Dox, offering new insights into potential strategies for enhancing cancer treatment efficacy.

## Effect of Dox-loaded AS-T9/U4_MH on hTERT and vimentin expression

The expression of hTERT and vimentin plays a crucial role in a process of epithelial-mesenchymal transition (EMT), which is pivotal in cancer progression by transforming epithelial cells into a mesenchymal state, thus significantly contributing to tumor development [53–55]. To explore how T9/U4 ASO regulates hTERT expression, we evaluated the hTERT levels in SW480 cells. Decreased hTERT levels led to the suppression of telomerase activity, which in turn benefitted by reducing cancer cell proliferation [56], enhancing apoptosis [57], and improving sensitivity to chemotherapy [58]. SW480 cells treated with AS-T9/U4_MH and Dox-loaded AS-T9/U4_MH showed a significant reduction in hTERT expression levels compared to cells treated with MHs lacking ASO (Fig 9A and 9B). These results suggested that T9/U4 ASO effectively downregulated hTERT levels in SW480 cells. Our results aligned with several studies indicating that T9/U4 modified with phosphorothioate (PS) had a potent effect on inhibiting telomerase activity in HL-60, A549-luc, and U-251 MG cells [35,59]. Moreover, the AS1411 aptamer and Dox in the MH contributed to lowering hTERT expression. The AS1411 aptamer naturally forms a G-quadruplex structure, effectively inhibiting telomerase activity by targeting telomeric G-quadruplexes and stabilizing those present in the hTERT promoter [60]. Dox was also reported to inhibit telomerase activity in SW480 cells [61], exhibiting a toxic effect that reduces both telomerase activity and hTERT expression [62]. Thus, loading Dox into AS-T9/U4_MH could further decrease hTERT level. These results indicated that T9/U4 ASO, AS1411 aptamer, and Dox acted synergistically to lower hTERT expression.

The vimentin expression level was further investigated. Treatment with both AS-T9/U4_MH and Dox-loaded AS-T9/U4_MH led to a decrease in vimentin expression in SW480

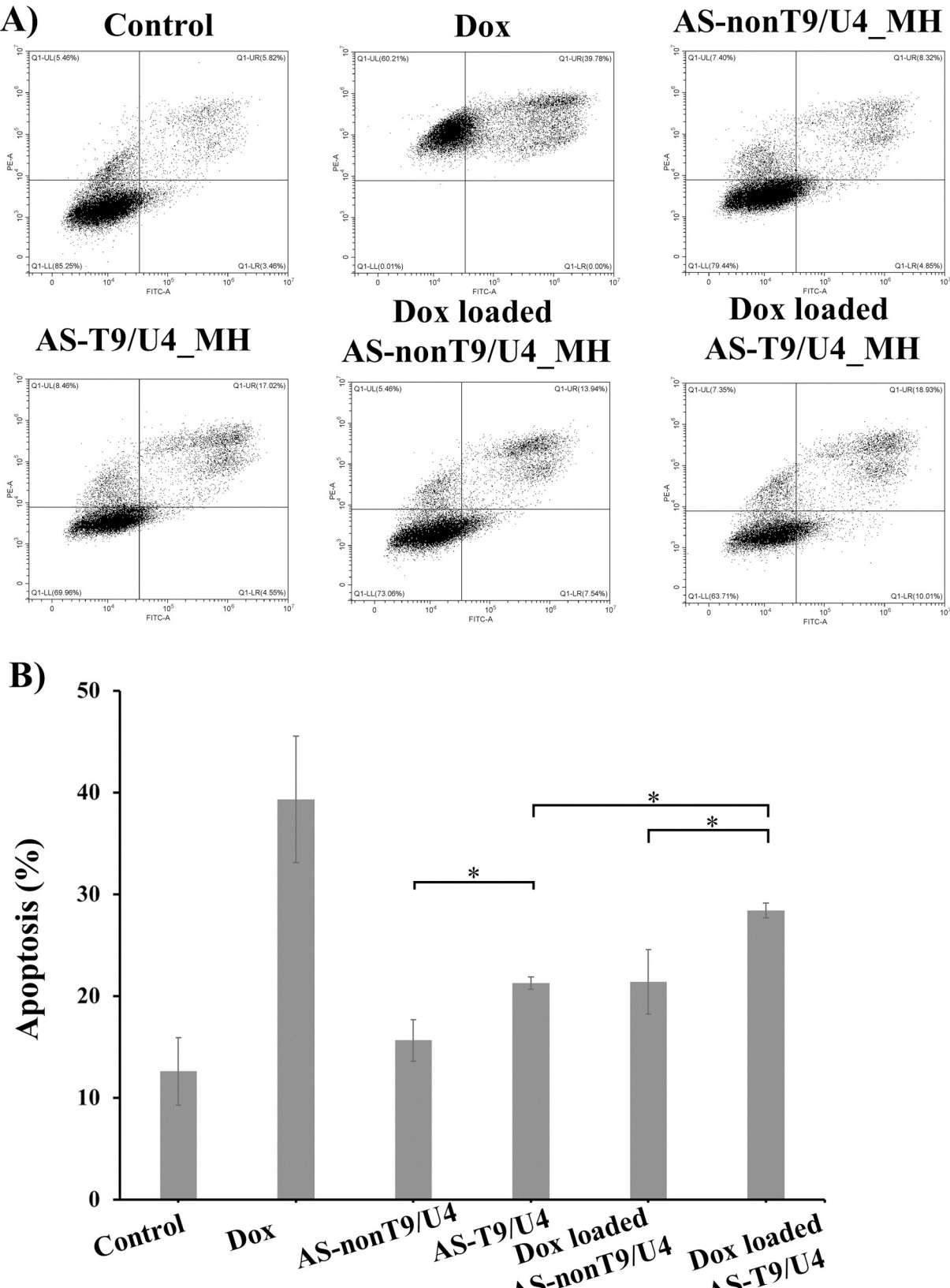

**Fig 8. Cell apoptosis of SW480 cells treated with 0.95 μM of Dox and 10 μM of AS-T9/U4_MH, AS-nonT9/U4_MH, Dox-loaded AS-T9/ U4_MH, and Dox-loaded AS-nonT9/U4_MH at 48 h, and no treatment as a control.** A) flow cytometry plot. B) Cell apoptosis count. The values are presented as means ±SD, n = 3, *P < 0.05.

cells compared to cells treated with MHs lacking the specific ASO (Fig 9A and 9C). These findings suggested that T9/U4 ASO effectively reduced vimentin levels through hTERT dysfunction. Additionally, the results indicated that Dox and the AS1411 aptamer slightly decreased vimentin expression in vitro. The reduced effect of Dox on vimentin expression in SW480 cells was also noted in a study on MFT-16 cells, which investigated the impact of vimentin on drug resistance. This study revealed that a mutant vimentin attached to mitochondria, enhancing cell membrane potential, consequently resulting in decreased Dox effectiveness [63]. As for the AS1411 aptamer, vimentin expression in SNU-761 hepatocellular carcinoma (HCC) remained unaltered after treatment with the aptamer, likely due to a key survival signaling pathway of HCC involving the PI3K/Akt or ERK1/2-MAPK pathway [64].

## Effect of Dox-loaded AS-T9/U4_MH on Bcl-2 and Bax expression

Bcl-2 and Bax play pivotal roles in apoptotic pathways, with Bcl-2 exhibiting anti-apoptotic properties and Bax known for its pro-apoptotic function [65]. Investigating the impact of our MHs on the expression levels of Bcl-2 and Bax in SW480 cells can provide crucial insights into related apoptotic pathways. Upon treatment with MHs, a notable decrease in Bcl-2 expression levels was observed in cells treated with free Dox, AS-T9/U4_MH, Dox-loaded AS-nonT9/ U4_MH, and Dox-loaded AS-T9/U4_MH formulations (Fig 10A and 10B). Notably, the MH formulation containing Dox, the aptamer, and the targeted ASO exhibited the most significant reduction in Bcl-2 expression. Dox facilitated apoptosis by upregulating the expression of Bax, caspase-8, and caspase-3, while concurrently downregulating Bcl-2 expression, as demonstrated in MCF-10F, MCF-7, and MDA-MB-231 breast cancer cell lines [66]. Interestingly, the AS1411 aptamer showed no effect on Bcl-2 expression, despite its reported inhibition of SW480 cell proliferation cells [34]. Strikingly, T9/U4 ASO markedly influenced Bcl-2

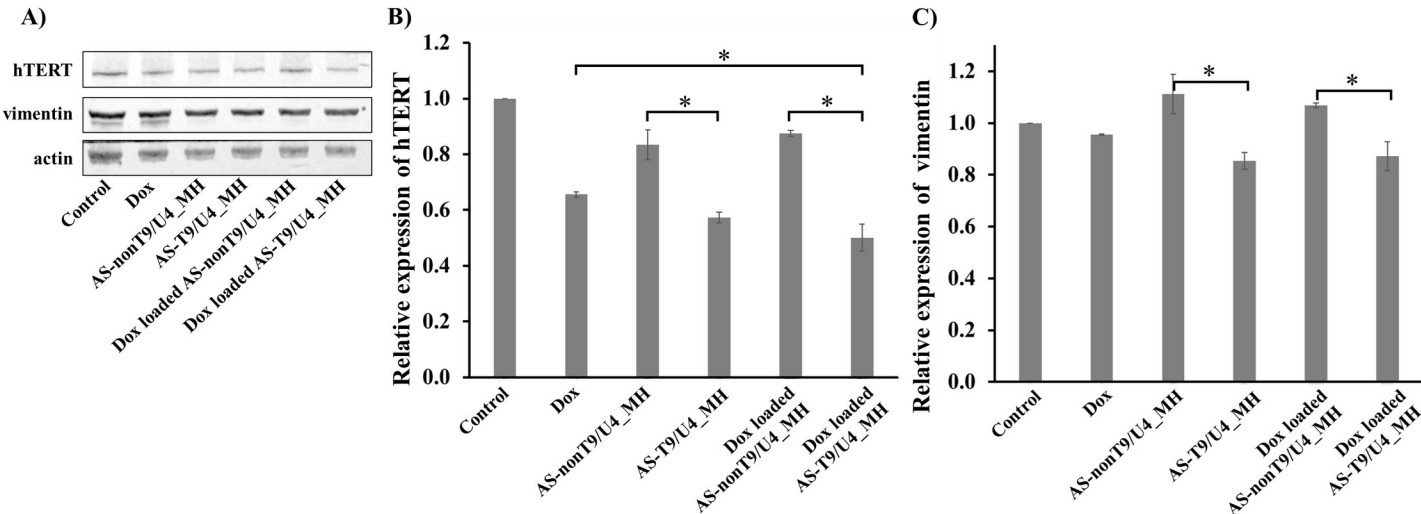

**Fig 9.** (A) Western blot analysis of hTERT and vimentin expression in SW480 cells treated with Dox, AS-T9/U4_MH, AS-nonT9/U4_MH, Dox-loaded AS-T9/U4_MH, and Dox-loaded AS-nonT9/U4_MH, and no treatment as a control. The relative expression of (B) hTERT and (C) vimentin. The values are presented as means ±SD, n = 3, *P < 0.05.

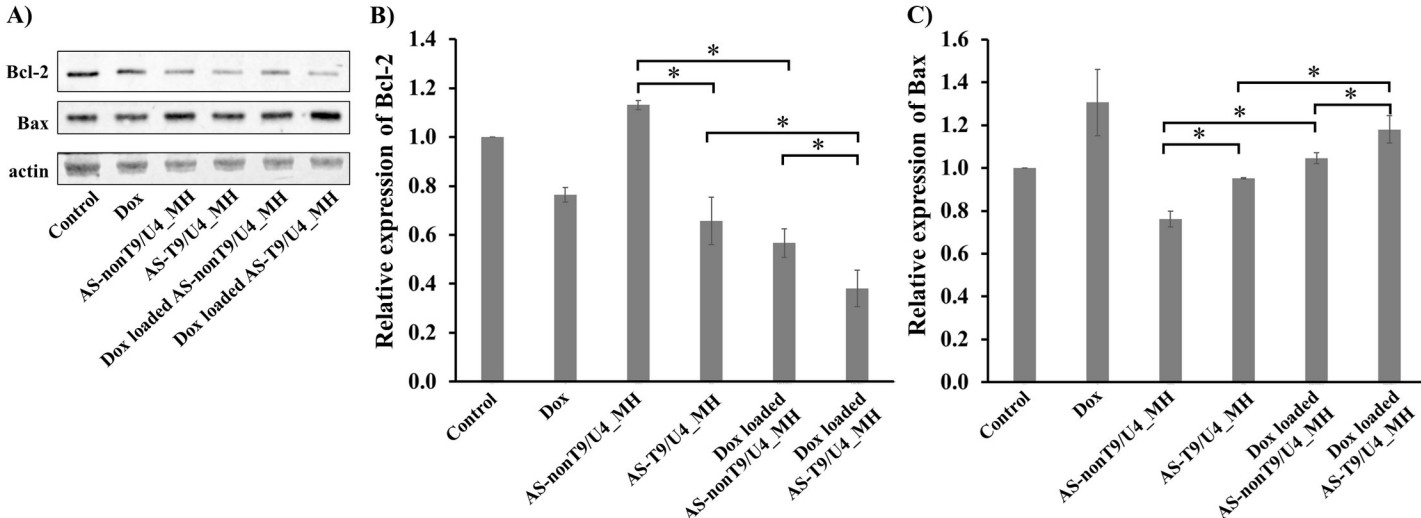

**Fig 10.** (A) Western blot analysis of Bcl-2 and Bax expression in SW480 cells treated with Dox, AS-T9/U4_MH, AS-nonT9/U4_MH, Dox-loaded AS-T9/U4_MH, and Dox-loaded AS-nonT9/U4_MH, and no treatment as a control. The relative expression of (B) Bcl-2 and (C) Bax. The values are presented as means ±SD, n = 3, *P < 0.05.

expression, emphasizing its efficacy in promoting cell apoptosis. Regarding Bax protein, the MH containing Dox and T9/U4 ASO notably increased its expression level (Fig 10A and 10C). Previous literature suggested that Dox induces apoptosis in MCF-7 breast cancer cells via the mitochondrial pathway by decreasing Bcl-xL expression and increasing Bax expression in a dose-dependent manner [67]. The expression patterns of both proteins indicated that Dox-loaded AS-T9/U4_MH affected this apoptotic pathway, with Dox and the ASO synergistically enhancing apoptosis in SW480 cells.

## Conclusions

This work accomplished following aspects. Gel electrophoresis confirmed the formation of MH, indicating successful preparation of AS-T9/U4_MH. The MH created was found to specifically target SW480 cells, as demonstrated by cell viability assays, fluorescence microscopy, and flow cytometry, while showing no significant impact on Caco-2 and CCD 841 CoN cells. AS-T9/U4_MH effectively served as a carrier for delivering ASO and Dox, leading to a synergistic effect observed in cell viability results. Interestingly, loading Dox into the DNA double helix resulted in reduced cytotoxicity. Furthermore, Dox-loaded AS-T9/U4_MH effectively downregulated hTERT and vimentin expression in SW480 cells, indicating its potential in inhibiting epithelial-mesenchymal transition (EMT) and tumor progression. Lastly, the impact of MHs on apoptotic pathways was investigated, revealing a significant decrease in Bcl-2 expression and increased Bax expression in SW480 cells treated with Dox-loaded AS-T9/U4_MH. This suggests the involvement of the mitochondrial pathway in apoptosis induction, with synergistic effects between Dox and T9/U4 ASO. Although the study employed in vitro cell models (SW480, Caco-2, and CCD 841 CoN) to evaluate the efficacy of the AS-T9/U4 molecular hybrid (MH), these models do not fully capture the complexity of the in vivo tumor microenvironment. Factors such as tumor heterogeneity, immune system interactions, and the presence of stromal components could influence the MH's performance in a clinical context. Therefore, further validation using animal models is necessary to confirm these findings. To enhance the translational potential of AS-T9/U4_MH, future research should prioritize in vivo studies to assess pharmacokinetics, biodistribution, tumor targeting, and overall

therapeutic efficacy, providing critical insights into the clinical feasibility of this molecular hybrid. Our findings demonstrate the potential of oligonucleotide hybridization in forming versatile macromolecular complexes with multiple functionalities. The success of this approach suggests that MH could serve as efficient drug delivery systems for cancer therapy.

## Supporting information

**S1 Fig. Dox intercalated using confocal microscope.** To assess the intercalation of Dox into dsDNA SW480 cells were seeded in 8-well chamber slide at density $9 \times 10^5$ and incubated 24 h. After, the cells were treated with 10 μM Dox-loaded AS-T9/U4_MH for 1.5 h. After treatment, the cells were washed twice with PBS and stained with DAPI to visualize the nuclei. Subsequently, the cells were imaged using a CLSM.
(TIF)

**S2 Fig. A standard curve of Dox.** Standard curve was created to evaluate the concentration of Dox remaining in the solution after intercalation. Dox solutions at various concentrations, including 0.95, 0.48, 0.24, 0.12, 0.06, 0.03, and 0.015 μM, were prepared. The fluorescence intensity at 590 nm was measured when excited at 480 nm using a Virokcan microplate reader.
(TIF)

**S3 Fig. Dox loading.** To determine Dox loading, the molar ratio of AS-T9/U4_MH to Dox was fixed at 1:0.095. Measurements were taken at different concentrations (5, 10, 15, and 20 μM) of AS-T9/U4_MH, with corresponding Dox concentrations of 0.475, 0.95, 1.9, and 3.8 μM, respectively. The loading efficiency (*LE*) was calculated using following equation: $LE = \frac{C_i - C_r}{C_i} x\ 100$ where $C_i$ is an initial Dox concentration, and $C_r$ is the remaining Dox concentration after intercalation.
(TIF)

**S4 Fig. AS-T9/U4_MH capacity for Dox loading.** To assess the capacity of AS-T9/U4_MH, 10 μM of AS-T9/U4_MH was incubated with varying concentrations of Dox (0, 0.95, 5, 10, 20, and 40 μM) in PBS solution at room temperature for 1.5 h. The fluorescence intensity of Dox was then measured using a Virokcan microplate reader.
(TIF)

**S5 Fig. Dox released.** The following experiment measured the amount of Dox released from AS-T9/U4_MH. Dox-loaded AS-T9/U4_MH was incubated in cell culture media at 37˚C. Samples were collected at 1, 3, 6, 12, 24, 48, and 72 hours, and the absorption was scanned over a wavelength range of 350 to 800 nm. The maximum absorption peak at 409 nm was used to calculate the percentage of Dox release. A concentration of 0.95 μM of Dox is considered 100%.
(TIF)

**S6 Fig. Stability of MH in the cell culture environment.** Electrophoresis results of Dox-loaded AS-T9/U4_MH after incubation in cell culture media, with samples collected at various time points. Lane M represents the DNA marker, lane 1 is the Dox-loaded AS-T9/U4_MH, and lanes 2 through 8 correspond to sampling times at 1 h, 3 h, 6 h, 12 h, 24 h, 48 h, and 72 h, respectively.
(TIF)

**S7 Fig. Trypan blue assay.** To evaluated the effect of Dox and T9/U4 ASO on SW480 cells proliferation, cells were seeded in 12-well plate at density $2 \times 10^6$ and incubated 24h. After that

the cells were treated with 10 µM of AS-T9/U4_MH, Dox-loaded nonAS-T9/U4_MH, Dox-loaded AS-nonT9/U4_MH and Dox-loaded AS-T9/U4_MH for 48 h. After treatment, the cells were collected and staned with trypan blue. Subsequently, the cells were imaged using a microscope (Nikon eclipse ts2r).
(TIF)

**S1 Raw images.**
(PDF)

## Author Contributions

**Conceptualization:** Boonchoy Soontornworajit.

**Data curation:** Kanpitcha Jiramitmongkon, Jiraporn Arunpanichlert.

**Formal analysis:** Kanpitcha Jiramitmongkon, Pichayanoot Rotkrua, Paisan Khanchaitit, Jiraporn Arunpanichlert, Boonchoy Soontornworajit.

**Funding acquisition:** Boonchoy Soontornworajit.

**Investigation:** Pichayanoot Rotkrua, Boonchoy Soontornworajit.

**Methodology:** Kanpitcha Jiramitmongkon, Boonchoy Soontornworajit.

**Project administration:** Boonchoy Soontornworajit.

**Supervision:** Pichayanoot Rotkrua, Boonchoy Soontornworajit.

**Writing – original draft:** Kanpitcha Jiramitmongkon, Boonchoy Soontornworajit.

**Writing – review & editing:** Kanpitcha Jiramitmongkon, Pichayanoot Rotkrua, Paisan Khanchaitit, Jiraporn Arunpanichlert, Boonchoy Soontornworajit.

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
