## [Decision Letter · Decision Letter 0]

20 Oct 2024

PONE-D-24-43470Multifunctional Molecular Hybrid for Targeted Colorectal Cancer Cells: Integrating Doxorubicin, AS1411 Aptamer, and T9/U4 ASOPLOS ONE

Dear Dr. Soontornworajit,

Thank you for submitting your manuscript to PLOS ONE. After careful consideration, we feel that it has merit but does not fully meet PLOS ONE’s publication criteria as it currently stands. Therefore, we invite you to submit a revised version of the manuscript that addresses the points raised during the review process.

As you can see from the comments, the reviewers felt that the scientific soundness of this study should be improved before accepting this work. I hope the specific comments from the reviewers will be useful for the major revision suggested by the reviewer. To help me expedite processing, please explicitly address the questions raised by the reviewers in your cover letter and also point out the changes made in the manuscript. I will go back to the reviewers for further input and advice before making any final decision on possible publication

We look forward to receiving your revised manuscript.

Kind regards,

Bing Xu, PhD

Academic Editor

PLOS ONE

Journal Requirements:

2. Thank you for stating the following financial disclosure: This study was supported by Thailand Science Research and Innovation Fundamental Fund (Contract No. TUFF 27/2567), and Thammasat University Research Unit in Innovation of Molecular Hybrid for Biomedical Application.  

3. Thank you for stating the following in the Acknowledgments Section of your manuscript: This study was supported by Thailand Science Research and Innovation Fundamental Fund (Contract No. TUFF 27/2567), and Thammasat University Research Unit in Innovation of Molecular Hybrid for Biomedical Application. In

addition, BS was financially supported for the travelling, and the participation of JCA 82nd 396 annual meeting by Thammasat University and the Japanese Cancer Association. KJ received scholarship from Faculty of Science and Technology, Thammasat University and National Nanotechnology Center (NANOTEC), National Science and Technology Development Agency(NSTDA). 

Please remove any funding-related text from the manuscript and let us know how you would like to update your Funding Statement. Currently, your Funding Statement reads as follows: This study was supported by Thailand Science Research and Innovation Fundamental Fund (Contract No. TUFF 27/2567), and Thammasat University Research Unit in Innovation of Molecular Hybrid for Biomedical Application. 

Reviewers' comments:

Reviewer's Responses to Questions

**Comments to the Author**

1. Is the manuscript technically sound, and do the data support the conclusions?

Reviewer #1: Yes

Reviewer #2: Partly

2. Has the statistical analysis been performed appropriately and rigorously? 

Reviewer #1: Yes

Reviewer #2: No

3. Have the authors made all data underlying the findings in their manuscript fully available?

Reviewer #1: Yes

Reviewer #2: Yes

4. Is the manuscript presented in an intelligible fashion and written in standard English?

Reviewer #1: Yes

Reviewer #2: Yes

5. Review Comments to the Author

Reviewer #1: This work studied the potential anticancer acitivity of a multifunctional molecular hybrid which integrates doxorubicin, AS1411 aptamer, and T9/U4 ASO. The general research idea and the experimental strategy are reasonable. However, the editting quality of manuscript were bad. For example, the sections of abstract and introduction are too verbose. Many well-known contents do not require detailed elaboration.

Reviewer #2: The manuscript by Boonchoy Soontornworajit et. al reported the design of a molecular hybrid (MH) containing doxorubicin, AS1411 aptamer, and T9/U4 ASO for the selective inhibition of nucleolin overexpressed colorectal cancer cells. The authors addressed most of the questions from the previous reviewers, but the significance of the design is still not well established because of the low anticancer efficacy. Therefore, I do not recommend accepting this paper until the advantages of the design are clearly demonstrated and some major issue get addressed.

1. The author stated that when molar ratio of AS-T9/U4 MH to Dox reaches 1:1, the fluorescence intensity reaches minimum, which is inconsistent with S4 Fig. In S4 Fig, 1:0.095 is the minimum, which is 10:1. And the calculation of loading efficiency is still unclear.

2. What is the ratio of T9/U4-HBS, AS1411-HBS, and a hybridization strand when making the molecular hybrid and how to determine the concentration? How to collect molecular hybrid after you making it? How to collect Dox loaded molecular hybrid?

3. Why did the author use 5 μM of MH in the flow cytometry part instead of using 10 μM as same as other experiment? Why did the authors use 10 μM, which is not effective according to the cell viability. The cell viability of SW480 being treated with AS-&9/U4_MH is greater than 70 %. The cell viability of SW480 being treated with Dox loaded AS-&9/U4_MH is greater than 60 %.

4. The great fluorescence increases of Caco-2 cells being treated with FAM-labeled nonAS-T9/U4_MH observed using flow cytometry is not consistent with fluorescence images. Please explain.

5. Could the author briefly explain the dissociation mechanism of Dox loaded MH inside cells?

6. PLOS authors have the option to publish the peer review history of their article (what does this mean?). If published, this will include your full peer review and any attached files.

Reviewer #1: No

Reviewer #2: No

---

## [Author Response · Author response to Decision Letter 0]

3 Dec 2024

Reviewer #1: This work studied the potential anticancer acitivity of a multifunctional molecular hybrid which integrates doxorubicin, AS1411 aptamer, and T9/U4 ASO. The general research idea and the experimental strategy are reasonable. However, the editing quality of manuscript were bad. For example, the sections of abstract and introduction are too verbose. Many well-known contents do not require detailed elaboration.

Response: As suggested, abstract and introduction are revised. (Line 22-24, 34-35, 38-40, 44-48, 52-53, 84-86)

Reviewer #2: 

The manuscript by Boonchoy Soontornworajit et. al reported the design of a molecular hybrid (MH) containing doxorubicin, AS1411 aptamer, and T9/U4 ASO for the selective inhibition of nucleolin overexpressed colorectal cancer cells. The authors addressed most of the questions from the previous reviewers, but the significance of the design is still not well established because of the low anticancer efficacy. Therefore, I do not recommend accepting this paper until the advantages of the design are clearly demonstrated and some major issue get addressed.

Response: The effect of AS-T9/U4_MH and Dox-loaded AS-T9/U4_MH on cell proliferation was evaluated using the MTS assay. However, the comparison between MH with and without Dox was not included in the previous manuscript submission. To address this, we analyzed and compared the cell viability levels in cells treated with AS-T9/U4_MH and Dox-loaded AS-T9/U4_MH, and the findings have been incorporated into the revised manuscript (Fig. 6). The results demonstrated a significant difference in cell viability between these two groups, highlighting the enhanced antiproliferative effect resulting from Dox incorporation in the MH (Line 146, 278-280, 300). To further emphasize the efficacy of the MH in suppressing cell proliferation, a trypan blue assay was performed and included in the revised manuscript (S7 Fig) (Line 601-605). The results revealed that SW480 cells treated with Dox-loaded AS-T9/U4 exhibited significantly greater antiproliferative effects compared to AS-T9/U4, Dox-loaded nonAS-T9/U4, and Dox-loaded AS-nonT9/U4. These findings are consistent with the MTS assay results. The trypan blue assay provided a clear distinction between live and dead cells based on membrane integrity, further supporting the effectiveness of the MH (Line 282-286).

1. The author stated that when molar ratio of AS-T9/U4 MH to Dox reaches 1:1, the fluorescence intensity reaches minimum, which is inconsistent with S4 Fig. In S4 Fig, 1:0.095 is the minimum, which is 10:1. And the calculation of loading efficiency is still unclear.

Response: From the manuscript, “Additionally, the Dox loading capacity of AS-T9/U4_MH was assessed, as shown in S4 Fig, revealing that as the concentration of Dox increased, the fluorescence intensity also rose. When the molar ratio of AS-T9/U4_MH to Dox was 1:1, the fluorescence intensity of Dox nearly reached its minimum, suggesting that the loading capacity of our MH was a 1:1 molar ratio.” These statements could confuse readers, as they describe how we estimated the Dox loading capacity based on the minimum detectable fluorescence signal. Following the reviewer's comments, we have clarified the ambiguous wording. “When the molar ratio of AS-T9/U4_MH to Dox was 1:1, the fluorescence intensity of Dox nearly reached the minimum detectable signal, suggesting that the loading capacity of our MH was a 1:1 molar ratio.” (Line 256)

In addition, we have carried out a standard curve of Dox fluorescence to determine Dox loading as follows. Standard curve was created to evaluate the concentration of Dox remaining in the solution after intercalation. Dox solutions at various concentrations, including 0.95, 0.48, 0.24, 0.12, 0.06, 0.03, and 0.015 µM, were prepared. The fluorescence intensity at 590 nm was measured when excited at 480 nm using a Virokcan microplate reader. The fluorescence intensity of Dox was measured to evaluate drug loading into AS-T9/U4_MH using a calibration curve method (S2 and S3 Fig.). The loading efficiency (LE) was calculated using following equation: LE=(C_i-C_r)/C_i x 100 where Ci is an initial Dox concentration, and Cr is the remaining Dox concentration after intercalation. This information is included in the supporting information (Line 585-587) 

2. What is the ratio of T9/U4-HBS, AS1411-HBS, and a hybridization strand when making the molecular hybrid and how to determine the concentration? How to collect molecular hybrid after you making it? How to collect Dox loaded molecular hybrid?

Response: To prepare the molecular hybrid (MH), all sequences were used at a concentration of 10 µM, maintaining a 1:1:1 molar ratio of T9/U4-HBS, AS1411-HBS, and the hybridization strand. The gel image (Fig. 1) confirms a single band corresponding to the complete oligonucleotide formulation, indicating that the MH concentration was indeed 10 µM. For Dox-loaded MH preparations, Dox was incorporated at specified concentrations into the 10 µM MH, and these formulations were subsequently analyzed in our study.

3.1 Why did the author use 5 μM of MH in the flow cytometry part instead of using 10 μM as same as other experiment? 3.2 Why did the authors use 10 μM, which is not effective according to the cell viability. The cell viability of SW480 being treated with AS-T9/U4_MH is greater than 70 %. The cell viability of SW480 being treated with Dox loaded AS-T9/U4_MH is greater than 60 %.

Response: In the early stages of our study, we used 10 µM of MH in flow cytometry, which revealed non-specific binding in SW480 cells, as shown in Figure R1. To address this issue, the MH concentration was reduced to 5 µM for flow cytometry, which effectively minimized non-specific binding while still demonstrating the binding capability of the MH, as illustrated in Fig. 3 of the manuscript. For cell viability assays, we followed our previous publication [1] and used an MH concentration of 10 µM. The cell viability of SW480 cells treated with AS-T9/U4_MH and Dox-loaded AS-T9/U4_MH was approximately 70% and 60%, respectively, with a significant difference that highlights the cytotoxic effect of Dox in the MH (Figure R2). Furthermore, these viability levels exceeded the 50% benchmark for Dox at its IC50 concentration, demonstrating the MH's advantage in reducing Dox toxicity.

Figure R1. Flow cytometry histogram of SW480 and Caco-2 cells treated with 10 �M of FAM-labeled nonAS-T9/U4_MH and FAM-labeled AS-T9/U4_MH at 37 ºC for 1.5 h, and no treatment as a control.

Figure R2. The cell viability of SW480 after treated with 10 �M of AS-T9/U4_MH and Dox-loaded AS-T9/U4_MH at 37�C for 48h. *P < 0.05. The data are presented as means ± SD, n = 3.

[1.] Rotkrua P, Lohlamoh W, Watcharapo P, Soontornworajit B. A molecular hybrid comprising AS1411 and PDGF-BB aptamer, cholesterol, and doxorubicin for inhibiting proliferation of SW480 cells. Journal of Molecular Recognition. 2021;34(11):e2926. doi: https://doi.org/10.1002/jmr.2926.

4. The great fluorescence increases of Caco-2 cells being treated with FAM-labeled nonAS-T9/U4_MH observed using flow cytometry is not consistent with fluorescence images. Please explain.

Response: Several studies discuss the challenges and conflicts between results obtained from flow cytometry and fluorescence microscopy. While these two methods are complementary, discrepancies often arise due to their different approaches to cell analysis. Flow cytometry measures fluorescence intensity at a population level, providing quantitative data but lacking morphological context. In contrast, microscopy captures visual details and individual cell morphology but is limited in statistical robustness. For instance, flow cytometry might detect population-wide changes in fluorescence, while microscopy could highlight cell-specific variability [1, 2]. For example, Domínguez-Fandos et al., when comparing flow cytometry and fluorescence microscopy for analyzing human sperm, noted that various factors can influence the results obtained through flow cytometry. These factors include differences in the methods used to define the target population, apply gating strategies, and calculate the frequency of positive events [3]. In our findings, the inconsistency between microscopy images and flow cytometry histograms observed in Caco-2 cells might be due to differences in population sizes: approximately 1,000 cells analyzed via microscopy compared to 30,000 cells via flow cytometry. Additionally, fluorescence bleaching might significantly affect the microscopic assay.

References

1. Brestoff, J. R.; Frater, J. L. Contemporary Challenges in Clinical Flow Cytometry: Small Samples, Big Data, Little Time. The Journal of Applied Laboratory Medicine 2022, 7 (4), 931-944. DOI: 10.1093/jalm/jfab176.

2. W.L. Godfrey, D.M. Hill, J.A. Kilgore, G.M. Buller, J.A. Bradford, D.R. Gray, I. Clements, K. Oakleaf, J.J. Salisbury, M.J. Ignatius, and M.S. Janes. Complementarity of Flow Cytometry and Fluorescence Microscopy. Microsc Microanal 2005, 11, 246-247.

3. Domınguez-Fandos D, Camejo MI, Ballesca JL, Oliva R. Human sperm DNA fragmentation: correlation of TUNEL results as assessed by flow cytometry and optical microscopy. Cytometry Part A 2007, 71A, 1011–1018.

5. Could the author briefly explain the dissociation mechanism of Dox loaded MH inside cells?

Response: After cellular internalization, the Dox-loaded MH releases Dox molecules within SW480 cells, possibly driven by diffusion due to the relatively low intracellular Dox concentrations. Alternatively, the release may occur through the gradual degradation of the aptamer by lysosomal endonucleases following uptake. These mechanisms, as described in the literature, may work together to facilitate Dox dissociation. This explanation was included in the manuscript. (Line 275-278) 

References

Bagalkot V, Farokhzad OC, Langer R, Jon S (2006) An aptamer-doxorubicin physical conjugate as a novel targeted drug-delivery platform. Angew Chem Int Ed Engl 45(48):8149–8152.

---

## [Decision Letter · Decision Letter 1]

15 Dec 2024

PONE-D-24-43470R1Multifunctional Molecular Hybrid for Targeted Colorectal Cancer Cells: Integrating Doxorubicin, AS1411 Aptamer, and T9/U4 ASOPLOS ONE

Dear Dr. Soontornworajit,

Thank you for submitting your manuscript to PLOS ONE. After careful consideration, we feel that it has merit but does not fully meet PLOS ONE’s publication criteria as it currently stands. Therefore, we invite you to submit a revised version of the manuscript that addresses the points raised during the review process.

As you can see from the enclosed reviews, the reviewers find your manuscript potentially suitable for publication in PLoS One. However, one of them raised a specific issue that must be addressed before the final acceptance of your manuscript. Therefore, I am requesting that you submit a revised version of this manuscript to address the comments. To help me expedite processing, please explicitly address the questions raised by the reviewer in your cover letter and also indicate the changes made in the manuscript.

We look forward to receiving your revised manuscript.

Kind regards,

Bing Xu, PhD

Academic Editor

PLOS ONE

Journal Requirements:

Reviewers' comments:

Reviewer's Responses to Questions

**Comments to the Author**

1. If the authors have adequately addressed your comments raised in a previous round of review and you feel that this manuscript is now acceptable for publication, you may indicate that here to bypass the “Comments to the Author” section, enter your conflict of interest statement in the “Confidential to Editor” section, and submit your "Accept" recommendation.

Reviewer #2: All comments have been addressed

Reviewer #3: (No Response)

2. Is the manuscript technically sound, and do the data support the conclusions?

Reviewer #2: Partly

Reviewer #3: Partly

3. Has the statistical analysis been performed appropriately and rigorously? 

Reviewer #2: Yes

Reviewer #3: Yes

4. Have the authors made all data underlying the findings in their manuscript fully available?

Reviewer #2: Yes

Reviewer #3: No

5. Is the manuscript presented in an intelligible fashion and written in standard English?

Reviewer #2: Yes

Reviewer #3: Yes

6. Review Comments to the Author

Reviewer #2: The authors have adequately addressed most of the comments from previous reviewers. The overall concept and experimental approach are reasoned and sound. Therefore, I recommend this manuscript for acceptance in PLOS ONE after addressing a few minor issues.

1. Regarding the ratio, although the authors explained that at a 1:1 molar ratio of AS-T9/U4_MH to Dox, the fluorescence intensity of Dox nearly reached the minimum detectable signal, it is unclear why they continued to use 0.95 µM Dox and 10 µM MH in the experiments, resulting in a 1:10 ratio. Additionally, in Figure S3, to determine Dox loading, the molar ratio of MH to Dox was fixed at 1:0.095. The rationale for these choices needs further clarification.

2. In Figure S1, it is unclear why Dox is not released from MH into the cytosol. If Dox were released, fluorescence should be observed. The authors should provide an explanation or additional data to clarify this observation.

3. In Figure 6, there is an inconsistency between the colors of the columns and the corresponding captions. The authors should ensure that the colors and captions are accurately aligned to avoid confusion.

Reviewer #3: The authors failed to respond to comments previously raised and the quality of this work still fall behind the criteria of PLOS ONE. Thus, I recommended the rejection of this work.

7. PLOS authors have the option to publish the peer review history of their article (what does this mean?). If published, this will include your full peer review and any attached files.

Reviewer #2: No

Reviewer #3: No

---

## [Author Response · Author response to Decision Letter 1]

26 Dec 2024

Reviewer #2: The authors have adequately addressed most of the comments from previous reviewers. The overall concept and experimental approach are reasoned and sound. Therefore, I recommend this manuscript for acceptance in PLOS ONE after addressing a few minor issues.

1. Regarding the ratio, although the authors explained that at a 1:1 molar ratio of AS-T9/U4_MH to Dox, the fluorescence intensity of Dox nearly reached the minimum detectable signal, it is unclear why they continued to use 0.95 µM Dox and 10 µM MH in the experiments, resulting in a 1:10 ratio. Additionally, in Figure S3, to determine Dox loading, the molar ratio of MH to Dox was fixed at 1:0.095. The rationale for these choices needs further clarification.

Response: The MH formulations presented in this study were designed with two main objectives. First, to evaluate the capability of MHs in cellular assays, including microscopy, flow cytometry, and cell viability tests. For these experiments, the Dox concentration was set at its IC50 value of 0.95 µM, while the MH concentration was maintained at 10 µM, based on findings from our previous publication [1]. Second, to assess Dox loading, fluorescence spectroscopy was used to investigate the fluorescence quenching of Dox at varying MH ratios, with the results shown in Figure S3.

 [1.] Rotkrua P, Lohlamoh W, Watcharapo P, Soontornworajit B. A molecular hybrid comprising AS1411 and PDGF-BB aptamer, cholesterol, and doxorubicin for inhibiting proliferation of SW480 cells. Journal of Molecular Recognition. 2021;34(11):e2926. doi: https://doi.org/10.1002/jmr.2926.

2. In Figure S1, it is unclear why Dox is not released from MH into the cytosol. If Dox were released, fluorescence should be observed. The authors should provide an explanation or additional data to clarify this observation.

Response: To evaluate the intercalation of Dox into dsDNA, SW480 cells were treated with Dox-loaded AS-T9/U4_MH for 1.5 hours and subsequently visualized using CLSM. As shown in Figure S1, the results confirm the complete intercalation of Dox within MH. The cellular imaging aligns with the Dox release experiment (Figure S5), which demonstrated that only about 5% of Dox was released from MH after 1 hour. This minimal amount of Dox is likely insufficient to be detected inside the cells using fluorescence microscopy. This explanation was included in the manuscripts (line 260-262)

3. In Figure 6, there is an inconsistency between the colors of the columns and the corresponding captions. The authors should ensure that the colors and captions are accurately aligned to avoid confusion.

Response: The colors in Figure 6 were adjusted to match its caption.

Reviewer #3: The authors failed to respond to comments previously raised and the quality of this work still fall behind the criteria of PLOS ONE. Thus, I recommended the rejection of this work.

Response: We appreciate the reviewers' comments.

---

## [Editor Report · Decision Letter 2]

2 Jan 2025

Multifunctional Molecular Hybrid for Targeted Colorectal Cancer Cells: Integrating Doxorubicin, AS1411 Aptamer, and T9/U4 ASO

PONE-D-24-43470R2

Dear Dr. Soontornworajit,

We’re pleased to inform you that your manuscript has been judged scientifically suitable for publication and will be formally accepted for publication once it meets all outstanding technical requirements.

Kind regards,

Bing Xu, PhD

Academic Editor

PLOS ONE
---

## [Editor Report · Acceptance letter]

17 Jan 2025

PONE-D-24-43470R2 

PLOS ONE

Dear Dr. Soontornworajit, 

I'm pleased to inform you that your manuscript has been deemed suitable for publication in PLOS ONE. Congratulations! Your manuscript is now being handed over to our production team.

Kind regards, 

on behalf of

Dr. Bing Xu 

Academic Editor

PLOS ONE